# ONLINE POLICY OPTIMIZATION FOR ROBUST MDP

## ABSTRACT

Reinforcement learning (RL) has exceeded human performance in many synthetic settings such as video games and Go. However, real-world deployment of end-to-end RL models is less common, as RL models can be very sensitive to slight perturbation of the environment. The robust Markov decision process (MDP) framework—in which the transition probabilities belong to an uncertainty set around a nominal model—provides one way to develop robust models. While previous analysis shows RL algorithms are effective assuming access to a generative model, it remains unclear whether RL can be efficient under a more realistic online setting, which requires a careful balance between exploration and exploitation. In this work, we consider online robust MDP by interacting with an unknown nominal system. We propose a robust optimistic policy optimization algorithm that is provably efficient. To address the additional uncertainty caused by an adversarial environment, our model features a new optimistic update rule derived via Fenchel conjugates. Our analysis establishes the first regret bound for online robust MDPs.

## 1 INTRODUCTION

The rapid progress of reinforcement learning (RL) algorithms enables trained agents to navigate around complicated environments and solve complex tasks. The standard reinforcement learning methods, however, may fail catastrophically in another environment, even if the two environments only differ slightly in dynamics (Farebrother et al., 2018; Packer et al., 2018; Cobbe et al., 2019; Song et al., 2019; Raileanu & Fergus, 2021). In practical applications, such mismatch of environment dynamics are common and can be caused by a number of reasons, e.g., model deviation due to incomplete data, unexpected perturbation and possible adversarial attacks. Part of the sensitivity of standard RL algorithms stems from the formulation of the underlying Markov decision process (MDP). In a sequence of interactions, MDP assumes the dynamic to be unchanged, and the trained agent to be tested on the same dynamic thereafter.

To model the potential mismatch between system dynamics, the framework of robust MDP is introduced to account for the uncertainty of the parameters of the MDP (Satia & Lave Jr, 1973; White III & Eldeib, 1994; Nilim & El Ghaoui, 2005; Iyengar, 2005). Under this framework, the dynamic of an MDP is no longer fixed but can come from some uncertainty set, such as the rectangular uncertainty set, centered around a nominal transition kernel. The agent sequentially interacts with the nominal transition kernel to learn a policy, which is then evaluated on the worst possible transition from the uncertainty set. Therefore, instead of searching for a policy that may only perform well on the nominal transition kernel, the objective is to find the worst-case best-performing policy. This can be viewed as a dynamical zero-sum game, where the RL agent tries to choose the best policy while nature imposes the worst possible dynamics. Intrinsically, solving the robust MDPs involves solving a max-min problem, which is known to be challenging for efficient algorithm designs.

More specifically, if a generative model (also known as a simulator) of the environment or a suitable offline dataset is available, one could obtain a $\epsilon$-optimal robust policy with $\tilde{O}(\epsilon^{-2})$ samples under a rectangular uncertainty set (Qi & Liao, 2020; Panaganti & Kalathil, 2022; Wang & Zou, 2022; Ma et al., 2022). Yet the presence of a generative model is stringent to fulfill for real applications. In a more practical online setting, the agent sequentially interacts with the environment and tackles the exploration-exploitation challenge as it balances between exploring the state space and exploiting the high-reward actions. In the robust MDP setting, previous sample complexity results cannot

directly imply a sublinear regret in general Dann et al. (2017) and so far no asymptotic result is available. A natural question then arises:

*Can we design a robust RL algorithm that attains sublinear regret under robust MDP with rectangular uncertainty set?*

In this paper, we answer the above question affirmatively and propose the first policy optimization algorithm for robust MDP under a rectangular uncertainty set. One of the challenges for deriving a regret guarantee for robust MDP stems from its adversarial nature. As the transition dynamic can be picked adversarially from a predefined set, the optimal policy may be randomized (Wiesemann et al., 2013). This is in contrast with conventional MDPs, where there always exists a deterministic optimal policy, which can be found with value-based methods and a greedy policy (e.g. UCB-VI algorithms). Bearing this observation, we resort to policy optimization (PO)-based methods, which directly optimize a stochastic policy in an incremental way.

With a stochastic policy, our algorithm explores robust MDPs in an optimistic manner. To achieve this robustly, we propose a carefully designed bonus function via the dual conjugate of the robust bellman equation. This quantifies both the uncertainty stemming from the limited historical data and the uncertainty of the MDP dynamic. In the episodic setting of robust MDPs, we show that our algorithm attains sublinear regret $O(\sqrt{K})$ for both $(s, a)$ and $s$-rectangular uncertainty set, where $K$ is the number of episodes. In the case where the uncertainty set contains only the nominal transition model, our results recover the previous regret upper bound of non-robust policy optimization (Shani et al., 2020). Our result achieves the first provably efficient regret bound in the online robust MDP problem, as shown in Table 1. We further validated our algorithm with experiments.

**Table 1:** Comparisons of previous results and our results, where $S, A$ are the size of the state space and action space, $H$ is the length of the horizon, $K$ is the number of episodes, $\rho$ is the radius of the uncertainty set and $\epsilon$ is the level of suboptimality. We shorthand $\iota = \log(SAH^2K^{3/2}(1+\rho))$. The regret upper bound by Panaganti & Kalathil (2022) are obtained through converting their sample complexity results and the sample complexity result for our work is converted through our regret bound. We use "GM" to denote the requirement of a generative model. The superscript * stands for results obtained via batch-to-online conversion. The reference to the previous works are [A]: Panaganti & Kalathil (2022), [B]: Wang & Zou (2021), [C]: Badrinath & Kalathil (2021), [D]: Yang et al. (2021).

| | Algorithm | Requires | Rectangular | Regret | Sample Complexity |
|---|---|---|---|---|---|
| [A] | Value based | GM | $(s, a)$ | $O\left(K^{\frac{2}{3}} H^{\frac{5}{3}} S^{\frac{2}{3}} A^{\frac{1}{3}}\right)^{*}$ | $O\left(\frac{H^4 S^2 A}{\epsilon^2}\right)$ |
| [B] | Value based | - | $(s, a)$ | NA | Asymptotic |
| [C] | Policy based | - | $(s, a)$ | NA | Asymptotic |
| [D] | Value based | GM | $(s, a)$ | NA | $\tilde{O}\left(\frac{H^4 S^2 A (2+\rho)^2}{\rho^2 \epsilon^2}\right)$ |
| | | | $s$ | NA | $\tilde{O}\left(\frac{H^4 S^2 A^2 (2+\rho)^2}{\rho^2 \epsilon^2}\right)$ |
| **Ours** | Policy based | - | $(s, a)$ | $O\left(SH^2\sqrt{AK\iota}\right)$ | $O\left(\frac{H^4 S^2 A \iota}{\epsilon^2}\right)$ |
| | | | $s$ | $O\left(SA^2 H^2 \sqrt{K\iota}\right)$ | $O\left(\frac{H^4 S^2 A^4 \iota}{\epsilon^2}\right)$ |

## 2 RELATED WORK

**RL with robust MDP** Different from conventional MDPs, robust MDPs allow the transition kernel to take values from an uncertainty set. The objective in robust MDPs is to learn an optimal robust policy that maximizes the worst-case value function. When the exact uncertainty set is known, this can be solved through dynamic programming methods (Iyengar, 2005; Nilim & El Ghaoui, 2005; Mannor et al., 2012). Yet knowing the exact uncertainty set is a rather stringent requirement for most real applications. If one has access to a generative model, several model-based reinforcement learning methods are proven to be statistically efficient. With the different characterization of the uncertainty set, these methods can enjoy a sample complexity of $O(1/\epsilon^2)$ for an $\epsilon$-optimal robust

value function (Panaganti & Kalathil, 2022; Yang et al., 2021). Similar results can also be achieved if an offline dataset is present, for which previous works Qi & Liao (2020); Zhou et al. (2021); Kallus et al. (2022); Ma et al. (2022) show the $O(1/\epsilon^2)$ sample complexity for an $\epsilon$-optimal policy. In addition, Liu et al. (2022) proposed distributionally robust policy Q-learning, which solves for the asymptotically optimal Q-function.

In the case of online RL, the only results available are asymptotic. In the case of discounted MDPs, Wang & Zou (2021); Badrinath & Kalathil (2021) study the policy gradient method and show an $O(\epsilon^{-3})$ convergence rate for an alternative learning objective (a smoothed variant), which could be equivalent to the original policy gradient objective in an asymptotic regime. These results in sample complexity and asymptotic regimes in general cannot imply sublinear regret in robust MDPs (Dann et al., 2017).

**RL with adversarial MDP**    Another line of works characterizes the uncertainty of the environment through the adversarial MDP formulation, where the environmental parameters can be adversarially chosen without restrictions. This problem is proved to be NP-hard to obtain a low regret (Even-Dar et al., 2004). Several works study the variant where the adversarial could only modify the reward function, while the transition dynamics of the MDP remain unchanged. In this case, it is possible to obtain policy-based algorithms that are efficient with a sublinear regret (Rosenberg & Mansour, 2019; Jin & Luo, 2020; Jin et al., 2020; Shani et al., 2020; Cai et al., 2020). On a separate vein, it investigates the setting where the transition is only allowed to be adversarially chosen for $C$ out of the $K$ total episodes. A regret of $O(C^2 + \sqrt{K})$ are established thereafter (Lykouris et al., 2021; Chen et al., 2021b; Zhang et al., 2022).

**Non-robust policy optimization**    The problem of policy optimization has been extensively investigated under non-robust MDPs (Neu et al., 2010; Cai et al., 2020; Shani et al., 2020; Wu et al., 2022; Chen et al., 2021a). The proposed methods are proved to achieve sublinear regret. The methods are also closely related to empirically successful policy optimization algorithms in RL, such as PPO Schulman et al. (2017) and TRPO Schulman et al. (2015).

## 3    ROBUST MDP AND UNCERTAINTY SETS

In this section, we describe the formal setup of robust MDP. We start with defining some notations.

**Robust Markov decision process**    We consider an episodic finite horizon robust MDP, which can denoted by a tuple $\mathcal{M} = \langle \mathcal{S}, \mathcal{A}, H, \{\mathcal{P}_h\}_{h=1}^H, \{r\}_{h=1}^H \rangle$. Here $\mathcal{S}$ is the state space, $\mathcal{A}$ is the action space, $\{r\}_{h=1}^H$ is the time-dependent reward function, and $H$ is the length of each episode. Instead of a fixed time-dependent uncertainty kernels, the transitions of the robust MDP is governed by kernels that are within a time-dependent uncertainty set $\{\mathcal{P}_h\}_{h=1}^H$, $i.e.$, time-dependent transition $P_h \in \mathcal{P}_h \subseteq \Delta_{\mathcal{S}}$ at time $h$.

The uncertainty set $\mathcal{P}$ is constructed around a nominal transition kernel $\{P_h^o\}$, and all transition dynamics within the set are close to the nominal kernel with a distance metric of one's choice. Different from an episodic finite-horizon non-robust MDP, the transition kernel $P$ may not only be time-dependent but may also be chosen (even adversarially) from a specified time-dependent uncertainty set $\mathcal{P}$. We consider the case where the rewards are stochastic. This is, on state-action $(s, a)$ at time $h$, the immediate reward is $R_h(s, a) \in [0, 1]$, which is drawn i.i.d from a distribution with expectation $r_h(s, a)$. With the described setup of robust MDPs, we now define the policy and its associated value.

**Policy and robust value function**    A time-dependent policy $\pi$ is defined as $\pi = \{\pi_h\}_{h=1}^H$, where each $\pi_h$ is a function from $\mathcal{S}$ to the probability simplex over actions, $\Delta(\mathcal{A})$. If the transition kernel is fixed to be $P$, the performance of a policy $\pi$ starting from state $s$ at time $h$ can be measured by its value function, which is defined as

$$V_h^{\pi, P}(s) = \mathbb{E}_{\pi, P} \left[ \sum_{h'=h}^H r_{h'}(s_{h'}, a_{h'}) \mid s_h = s \right].$$

In robust MDP, the robust value function instead measures the performance of $\pi$ under the worst possible choice of transition $P$ within the uncertainty set. Specifically, the value and the Q-value function of a policy given the state action pair $(s, a)$ at step $h$ are defined as

$$V_h^\pi(s) = \min_{\{P_h\} \in \{\mathcal{P}_h\}} V_h^{\pi, \{P\}}(s),$$

$$Q_h^\pi(s, a) = \min_{\{P_h\} \in \{\mathcal{P}_h\}} \mathbb{E}_{\pi, \{P\}} \left[ \sum_{h'=h}^{H} r_h(s_{h'}, a_{h'}) \mid (s_h, a_h) = (s, a) \right].$$

The optimal value function is defined to be the best possible value attained by a policy

$$V_h^*(s) = \max_\pi V_h^\pi(s) = \max_\pi \min_{\{P_h\} \in \{\mathcal{P}_h\}} V_h^{\pi, \{P\}}(s).$$

The optimal policy is then defined to be the policy that attains the optimal value.

**Robust Bellman equation**  Similar to non-robust MDP, robust MDP has the following robust bellman equation, which characterizes a relation to the robust value function (Ho et al., 2021; Yang et al., 2021).

$$Q_h^\pi(s, a) = r(s, a) + \sigma_{\mathcal{P}_h}(V_{h+1}^\pi)(s, a), \quad V_h^\pi(s) = \langle Q_h^\pi(s, \cdot), \pi_h(\cdot, s) \rangle,$$

where

$$\sigma_{\mathcal{P}_h}(V_{h+1}^\pi)(s, a) = \min_{P_h \in \mathcal{P}_h} P_h(\cdot \mid s, a) V_{h+1}^\pi, \quad P_h(\cdot \mid s, a) V = \sum_{s' \in \mathcal{S}} P_h(s' \mid s, a) V(s'). \quad (1)$$

Without additional assumptions on the uncertainty set, the optimal policy and value of the robust MDP are in general NP-hard to solve (Wiesemann et al., 2013). One of the most commonly assumptions that make solving optimal value feasible is the rectangular assumption (Iyengar, 2005; Wiesemann et al., 2013; Badrinath & Kalathil, 2021; Yang et al., 2021; Panaganti & Kalathil, 2022).

**Rectangular uncertainty sets**  To limit the level of perturbations, we assume that the transition kernels is close to the nominal transition measured via $\ell_1$ distance. We consider two cases.

The $(s, a)$-rectangular assumption assumes that the uncertain transition kernel within the set takes value independently for each $(s, a)$. We further use $\ell_1$ distance to characterize the $(s, a)$-rectangular set around a nominal kernel with a specified level of uncertainty.

**Definition 3.1** ($(s, a)$-rectangular uncertainty set Iyengar (2005); Wiesemann et al. (2013)). *For all time step $h$ and with a given state-action pair $(s, a)$, the $(s, a)$-rectangular uncertainty set $\mathcal{P}_h(s, a)$ is defined as*

$$\mathcal{P}_h(s, a) = \{\|P_h(\cdot \mid s, a) - P_h^o(\cdot \mid s, a)\|_1 \le \rho, P_h(\cdot \mid s, a) \in \Delta(\mathcal{S})\},$$

*where $P_h^o$ is the nominal transition kernel at $h$, $P_h^o(\cdot \mid s, a) > 0, \forall (s, a) \in \mathcal{S} \times \mathcal{A}$, $\rho$ is the level of uncertainty and $\Delta(\mathcal{S})$ denotes the probability simplex over the state space $\mathcal{S}$.*

With the $(s, a)$-rectangular set, it is shown that there always exists an optimal policy that is deterministic Wiesemann et al. (2013).

One way to relax the $(s, a)$-rectangular assumption is to instead let the uncertain transition kernels within the set take value independent for each $s$ only. This characterization is then more general and its solution gives a stronger robustness guarantee.

**Definition 3.2** ($s$-rectangular uncertainty set Wiesemann et al. (2013)). *For all time step $h$ and with a given state $s$, the $s$-rectangular uncertainty set $\mathcal{P}_h(s)$ is defined as*

$$\mathcal{P}_h(s) = \left\{ \sum_{a \in \mathcal{A}} \|P_h(\cdot \mid s, a) - P_h^o(\cdot \mid s, a)\|_1 \le A\rho, P_h(\cdot \mid s, \cdot) \in \Delta(\mathcal{S})^{\mathcal{A}} \right\},$$

*where $P_h^o$ is the nominal transition kernel at $h$, $P_h^o(\cdot \mid s, a) > 0, \forall (s, a) \in \mathcal{S} \times \mathcal{A}$, $\rho$ is the level of uncertainty, and $\Delta(\mathcal{S})$ denotes the probability simplex over the state space $\mathcal{S}$.*

Different from the $(s, a)$-rectangular assumption, which guarantees the existence of a deterministic optimal policy, the optimal policy under $s$-rectangular set may need to be randomized (Wiesemann et al., 2013). We also remark that the requirement of $P_h^o(\cdot \mid s, a) > 0$ is mostly for technical convenience.

Equipped with the characterization of the uncertainty set, we now describe the learning protocols and the definition of regret under the robust MDP.

**Learning protocols and regret**   We consider a learning agent repeatedly interacts with the environment in an episodic manner, over $K$ episodes. At the start of each episode, the learning agent picks a policy $\pi_k$ and interacts with the environment while executing $\pi_k$. Without loss of generality, we assume the agents always start from a fixed initial state $s$. The performance of the learning agent is measured by the cumulative regret incurred over the $K$ episodes. Under the robust MDP, the cumulative regret is defined to be the cumulative difference between the robust value of $\pi_k$ and the robust value of the optimal policy,

$$\text{Regret}(K) = \sum_{k=1}^{K} V_1^*(s_1^k) - V_1^{\pi_k}(s_1^k),$$

where $s_1^k$ is the initial state.

We highlight that the transition of the states in the learning process is specified by the nominal transition kernel $\{P_h^o\}_{h=1}^H$, though the agent only has access to the nominal kernel in an online manner. We remark that if the agent is asked to interact with a potentially adversarially chosen transition from an arbitrary uncertainty set, the learning problem is NP-hard Even-Dar et al. (2004).

One practical motivation for this formulation could be as follows. The policy provider only sees feedback from the nominal system, yet she aims to minimize the regret for clients who refuse to share additional deployment details for privacy purposes.

## 4   ALGORITHM

Before we introduce our algorithm, we first illustrate the importance of taking uncertainty into consideration. With the robust MDP, one of the most naive methods is to directly train a policy with the nominal transition model. However, the following proposition shows an optimal policy under the nominal policy can be arbitrarily bad in the worst-case transition (even worse than a random policy).

**Claim 4.1** (Suboptimality of non-robust optimal policy). *There exists a robust MDP $\mathcal{M} = \langle \mathcal{S}, \mathcal{A}, \mathcal{P}, r, H \rangle$ with uncertainty set $\mathcal{P}$ of uncertainty radius $\rho$, such that the non-robust optimal policy is $\Omega(1)$-suboptimal to the uniformly random policy.*

The proof of Proposition 4.1 is deferred to Appendix D. With the above-stated result, it implies the policy obtained with non-robust RL algorithms, can have arbitrarily bad performance when the dynamic mismatch from the nominal transition. Therefore, we present the following robust optimistic policy optimization 1 to avoid this undesired result.

### 4.1   ROBUST OPTIMISTIC POLICY OPTIMIZATION

With the presence of the uncertainty set, the optimal policies may be all randomized (Wiesemann et al., 2013). In such cases, value-based methods may be insufficient as they usually rely on a deterministic policy. We thus resort to optimistic policy optimization methods Shani et al. (2020), which directly learn a stochastic policy.

Our algorithm performs policy optimization with empirical estimates and encourages exploration by adding a bonus to less explored states. However, we need to propose a new efficiently computable bonus that is robust to adversarial transitions. We achieve this via solving a sub-optimization problem derived from Fenchel conjugate. We present Robust Optimistic Policy Optimization (ROPO) in Algorithm 1 and elaborate on its design components.

To start, as our algorithm has no access to the actual reward and transition function, we use the following empirical estimator of the transition and reward:

$$\hat{r}_h^k(s,a) = \frac{\sum_{k'=1}^{k-1} R_h^{k'}(s,a) \mathbb{I}\left\{s_h^{k'} = s, a_h^{k'} = a\right\}}{N_h^k(s,a)},$$

$$\hat{P}_h^{o,k}(s,a,s') = \frac{\sum_{k'=1}^{k-1} \mathbb{I}\left\{s_h^{k'} = s, a_h^{k'} = a, s_{h+1}^{k'} = s'\right\}}{N_h^k(s,a)}, \tag{2}$$

where $N_h^k(s,a) = \max\left\{\sum_{k'=1}^{k-1} \mathbb{I}\left\{s_h^{k'} = s, a_h^{k'} = a\right\}, 1\right\}$ counts the number of visits to $(s,a)$.

**Optimistic Robust Policy Evaluation** In each episode, the algorithm estimates $Q$-values with an optimistic variant of the bellman equation. Specifically, to encourage exploration in the robust MDP, we add a bonus term $b_h^k(s,a)$, which compensates for the lack of knowledge of the actual reward and transition model as well as the uncertainly set, with order $b_h^k(s,a) = O\left(N_h^k(s,a)^{-1/2}\right)$.

$$\hat{Q}_h^k(s,a) = \min\left\{\hat{r}(s,a) + \sigma_{\hat{\mathcal{P}}_h}(\hat{V}_{h+1}^{\pi})(s,a) + b_h^k(s,a), H\right\}, \quad \hat{V}_h^k(s) = \left\langle \hat{Q}_h^k(s,\cdot), \pi_h^k(\cdot \mid s)\right\rangle.$$

Intuitively, the bonus term $b_h^k$ desires to characterize the optimism required for efficient exploration for both the estimation errors of $P$ and the robustness of $P$. It is hard to control the two quantities in their primal form because of the coupling between them. We propose the following procedure to address the problem.

Note that the key difference between our algorithm and standard policy optimization is that $\sigma_{\hat{\mathcal{P}}_h}(\hat{V}_{h+1}^{\pi})(s)$ requires solving an inner minimization (1). Through relaxing the constraints with Lagrangian multiplier and Fenchel conjugates, under $(s,a)$-rectangular set, the inner minimization problem can be reduced to a one-dimensional unconstrained convex optimization problem on $\mathbb{R}$ (Lemma 4).

$$\sup_{\eta} \eta - \frac{(\eta - \min_s \hat{V}_{h+1}^{\pi_k}(s))_+}{2}\rho - \sum_{s'} \hat{P}_h^o(s' \mid s,a)\left(\eta - \hat{V}_{h+1}^{\pi_k}(s')\right)_+. \tag{3}$$

The optimum of Equation (3) is then computed efficiently with bisection or sub-gradient methods. We note that while the dual form has been similarly used before under the presence of a generative model or with an offline dataset (Badrinath & Kalathil, 2021; Panaganti & Kalathil, 2022; Yang et al., 2021), it remains unclear whether it is effective for the online setting.

Similarly, in the case of $s$-rectangular set, the inner minimization problem is equivalent to a $A$-dimensional convex optimization problem.

$$\sup_{\eta} \sum_{a'} \eta_{a'} - \sum_{s',a'} \hat{P}_h^o(s' \mid s,a')\left(\eta_{a'} - \mathbb{I}\{a' = a\}\hat{V}_{h+1}^{\pi_k}(s')\right)_+$$
$$- \min_{s',a'} \frac{A\rho(\eta_{a'} - \mathbb{I}\{a' = a\}\hat{V}_{h+1}^{\pi_k}(s'))_+}{2}, \tag{4}$$

where $a \sim \pi_k(s)$.

In addition to reducing computational complexity, the dual form (Equation (3) and Equation (4)) decouples the uncertainty in estimation error and in robustness, as $\rho$ and $\hat{P}_h^o$ are in different terms. The exact form of $b_h^k$ is presented in the Equation (5) and (6).

**Policy Improvement Step** Using the optimistic $Q$-value obtained from policy evaluation, the algorithm improves the policy with a KL regularized online mirror descent step,

$$\pi_h^{k+1} \in \arg\max_{\pi} \beta\langle \nabla \hat{V}_h^{\pi_k}, \pi\rangle - \pi_h^k + D_{KL}(\pi||\pi_h^k),$$

where $\beta$ is the learning rate. Equivalently, the updated policy is given by the closed-form solution $\pi_h^{k+1}(a \mid s) = \frac{\pi_h^k \exp(\beta \hat{Q}_h^\pi(s,a))}{\sum_{a'} \exp(\beta \hat{Q}_h^\pi(s,a'))}$. An important property of policy improvement is to use a fundamental inequality (7) of online mirror descent presented in (Shani et al., 2020). We suspect that other online algorithms with sublinear regret could also be used in policy improvement.

In the non-robust case, this improvement step is also shown to be theoretically efficient (Shani et al., 2020; Wu et al., 2022). Many empirically successful policy optimization algorithms, such as PPO (Schulman et al., 2017) and TRPO Schulman et al. (2015), also take a similar approach to KL regularization for non-robust policy improvement. Putting everything together, the proposed algorithm is summarized in Algorithm 1.

---

**Algorithm 1** Robust Optimistic Policy Optimization (ROPO)

---

Input: learning rate $\beta$, bonus function $b_h^k$.
**for** $k = 1, \ldots, K$ **do**
    Collect a trajectory of samples by executing $\pi_k$.
    # Robust Policy Evaluation
    **for** $h = H, \ldots, 1$ **do**
        **for** $\forall (s,a) \in \mathcal{S} \times \mathcal{A}$ **do**
            Solve $\sigma_{\hat{\mathcal{P}}_h}(\hat{V}_{h+1}^{\pi})(s,a)$ according to Equation (3) for $(s,a)$-rectangular set
            or Equation (4) for $s$-rectangular set.
            $\hat{Q}_h^k(s,a) = \min\left\{ \hat{r}(s,a) + \sigma_{\hat{\mathcal{P}}_h}(\hat{V}_{h+1}^{\pi})(s,a) + b_h^k(s,a), H \right\}$.
        **end for**
        **for** $\forall s \in \mathcal{S}$ **do**
            $\hat{V}_h^k(s) = \left\langle \hat{Q}_h^k(s,\cdot), \pi_h^k(\cdot \mid s) \right\rangle$.
        **end for**
    **end for**
    # Policy Improvement
    **for** $\forall h, s, a \in [H] \times \mathcal{S} \times \mathcal{A}$ **do**
        $\pi_h^{k+1}(a \mid s) = \frac{\pi_h^k \exp(\beta \hat{Q}_h^{\pi}(s,a))}{\sum_{a'} \exp(\beta \hat{Q}_h^{\pi}(s,a'))}$.
    **end for**
    Update empirical estimate $\hat{r}$, $\hat{P}$ with Equation (2).
**end for**

---

## 5 THEORETICAL RESULTS

We are now ready to analyze the theoretical results of our algorithm under the uncertainly set.

### 5.1 RESULTS UNDER $(s,a)$-RECTANGULAR UNCERTAINTY SET

Equipped with Algorithm 1 and the bonus function described in Equation 5. We obtain the regret upper bound under $(s,a)$-rectangular uncertainty set described in the following Theorem.

**Theorem 1** (Regret under $(s,a)$-rectangular uncertainty set)**.** *With learning rate $\beta = \sqrt{\frac{2\log A}{H^2 K}}$ and bonus term $b_h^k$ as (5), with probability at least $1 - \delta$, the regret incurred by Algorithm 1 over $K$ episodes is bounded by $Regret(K) = O\left(H^2 S \sqrt{AK \log\left(SAH^2 K^{3/2}(1+\rho)/\delta\right)}\right)$.*

**Remark 5.1.** *When $\rho = 0$, the problem reduces to non-robust reinforcement learning. In such case our regret upper bound is $\tilde{O}\left(H^2 S\sqrt{AK}\right)$, which is in the same order of policy optimization algorithms for the non-robust case Shani et al. (2020).*

While we defer the detailed proof to the appendix A, we sketch and highlight the challenges in the proof below.

First, unlike policy optimization for non-robust MDP, classic lemmas such as the value difference lemma (Shani et al., 2020) can be no longer applied, because the adversarial transition kernel are policy dependent. Naively employing a recursive relation with respect to a fixed transition kernel in a similar way to the value difference lemma may lead to linear regret. To address this issue, we

propose the following decomposition,

$$V_h^*(s) - \hat{V}_h^{\pi_k}(s) \leq \mathbb{E}_{\pi_*} \left[ (r_h(s,a) - \hat{r}_h^k(s,a)) + (\sigma_{\mathcal{P}_h(s,a)}(\hat{V}_{h+1}^{\pi_k})(s,a) - \sigma_{\hat{\mathcal{P}}_h(s,a)}(\hat{V}_{h+1}^{\pi_k})(s,a)) - b_h^k(s,a) \right]$$

$$+ \mathbb{E}_{\pi_*} \left[ \sigma_{\mathcal{P}_h(s,a)}(V_{h+1}^*)(s,a) - \sigma_{\mathcal{P}_h(s,a)}(\hat{V}_{h+1}^{\pi_k})(s,a) \right]$$

$$+ \langle \hat{Q}_h^{\pi_k}(s,\cdot), \pi_*(\cdot \mid s) - \pi_k(\cdot \mid s) \rangle.$$

In particular, we perform a recursion conditioned on varying transition kernel $p_h(\cdot \mid s,a) = \arg\max_{P_h \in \mathcal{P}_h} P_h(\cdot \mid s,a)(\hat{V}_{h+1}^{\pi_k} - V_{h+1}^{\pi_k})$.

However, this introduces another problem. Maintaining optimism is hard as the expectation of each time step $h$ is taken with respect to a different transition kernel. To establish an optimism bonus for the uncertainty of the transition caused by limited interaction and the uncertainty set, we derive the dual formulation of inner optimization problem $\sigma_{\hat{\mathcal{P}}_{(s,a)}}(V)$ (3). This allows us to decouple the uncertainty and bound each source of uncertainty separately.

Notice that now the difference of $\sigma_{\hat{\mathcal{P}}_{(s,a)}}(V) - \sigma_{\mathcal{P}_{(s,a)}}(V)$ is only incurred by the difference in $\sum_{s'} P_h^o(s' \mid s,a) \left( \eta - \hat{V}_{h+1}^{\pi_k}(s') \right)_+$. We then show that $\eta$ must be bounded at its optimum by inspecting certain pivot points and by the convexity of the dual. When we have the desired bounds of $\eta$, applying Hoeffding's inequality with an $\epsilon$-net argument will yield the claimed regret bound.

Our algorithm and analysis techniques can also extend to other uncertainty sets, such as KL divergence constrained uncertainly set. We include the KL divergence result in Appendix C.

## 5.2 Results under $s$-rectangular uncertainty set

Beyond the $(s,a)$-rectangular uncertainty set, we also extends to $s$-rectangular uncertainty set (Definition 3.2). Recall that value-based methods do not extend to $s$-rectangular uncertainty set as there might not exist a deterministic optimal policy.

**Theorem 2** (Regret under $s$-rectangular uncertainty set). *With learning rate $\beta = \sqrt{\frac{2\log A}{H^2 K}}$ and bonus term $b_h^k$ as (6), with probability at least $1 - \delta$, the regret of Algorithm 1 is bounded by*

*$$Regret(K) = O\left( SA^2 H^2 \sqrt{K \log(SA^2 H^2 K^{3/2}(1+\rho)/\delta)} \right).$$*

**Remark 5.2.** *When $\rho = 0$, the problem reduces to non-robust reinforcement learning. In such case our regret upper bound is $\tilde{O}\left( SA^2 H^2 \sqrt{K} \right)$. Our result is the first theoretical result for learning a robust policy under $s$-rectangular uncertainty set, as previous results only learn the robust value function (Yang et al., 2021).*

The analysis and techniques used for Theorem 2 hold great similarity to those ones used for Theorem 1. The main difference is on bounding $\sigma_{\hat{\mathcal{P}}_h(s)}(\hat{V}_{h+1}^{\pi_k})(s,a) - \sigma_{\mathcal{P}_h(s)}(\hat{V}_{h+1}^{\pi_k})(s,a)$. We defer the detailed proof to the appendix B.

## 6 Empirical results

To validate our theoretical findings, we conduct a preliminary empirical analysis of our purposed robust policy optimization algorithm.

**Environment**   We conduct the experiments with the Gridworld environment, which is an early example of reinforcement learning from Sutton & Barto (2018). The environment is two-dimensional and is in a cell-like environment. Specifically, the environment is a $5 \times 5$ grid, where the agent starts from the upper left cell. The cells consist of three types, road (labeled with $o$), wall (labeled with $x$), or reward state (labeled with $+$). The agent can safely walk through the road cell but not the wall cell. Once the agent steps on the reward cell, it will receive a reward of 1, and it will receive no rewards otherwise. The goal of the agents is to collect as many rewards as possible within the allowed time.

| Start | o | o | o | o |
|---|---|---|---|---|
| o | x | o | o | o |
| o | o | x | o | o |
| o | o | o | x | o |
| o | o | o | o | + |

**Figure 1:** Example of the Gridworld environment.

The agent has four types of actions at each step, up, down, left, and right. After taking the action, the agent has a success probability of $p$ to move according to the desired direction, and with the remaining probability of moving to other directions.

**Experiment configurations**   To simulate the robust MDP, we create a nominal transition dynamic with success probability $p = 0.9$. The learning agent will interact with this nominal transition during training time and interact with a perturbed transition dynamic during evaluation. Under $(s, a)$-rectangular set, the transitions are perturbed against the direction is agent is directing with a constraint of $\rho$. Under $s$-rectangular set, the transitions are perturbed against the direction of the goal state. Figure 1 shows an example of our environment, where the perturbation caused some of the optimal policies under nominal transition to be sub-optimal under robust transitions. We denote the perturbed transition as robust transitions in our results. We implement our proposed robust policy optimization algorithm along with the non-robust variant of it Shani et al. (2020). The inner minimization of our Algorithm 1 is computed through its dual formulation for efficiency. Our algorithm is implemented with the rLberry framework (Domingues et al., 2021).

**Results**   We present results with $\rho = 0.1, 0.2, 0.3$ under $(s, a)$-rectangular set here in Figure 4. The results with $s$-rectangular sets are included in the appendix. We present the averaged cumulative rewards during evaluation. Regardless of the level of uncertainty, we observe that the robust variant of the policy optimization algorithm is more robust to dynamic changes as it is able to obtain a higher level of rewards than its non-robust variant.

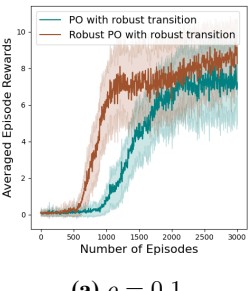 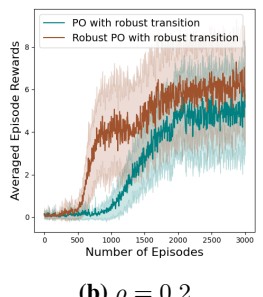 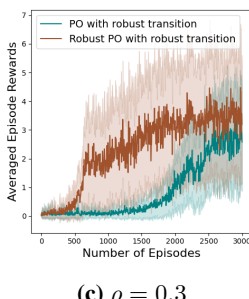

**(a)** $\rho = 0.1$    **(b)** $\rho = 0.2$    **(c)** $\rho = 0.3$

**Figure 2:** Cumulative rewards obtained by robust and non-robust policy optimization on robust transition with different level of uncertainty $\rho = 0.1, 0.2, 0.3$ under $\ell_1$ distance, $(s, a)$-rectangular set.

## 7   CONCLUSION AND FUTURE DIRECTIONS

In this paper, we studied the problem of regret minimization in robust MDP with a rectangular uncertainty set. We proposed a robust variant of optimistic policy optimization, which achieves sub-linear regret in all uncertainty sets considered. Our algorithm delicately balances the exploration-exploitation trade-off through a carefully designed bonus term, which quantifies not only the uncertainty due to the limited observations but also the uncertainty of robust MDPs. Our results are the first regret upper bounds in robust MDPs as well as the first non-asymptotic results in robust MDPs without access to a generative model.

For future works, while our analysis achieves the same bound as the policy optimization algorithm in Shani et al. (2020) when the robustness level $\rho = 0$, we suspect some technical details could be improved. For example, we required $P_h^o$ to be positive for any $s, a$ so that we could do a change of variable to form an efficiently solvable Fenchel dual. However, the actual positive value gets canceled out later and does not show up in the bound, suggesting that the strictly positive assumption might be an artifact of analysis.

Furthermore, our work could also be extended in several directions. One is to consider other characterization of uncertainty sets, such as the Wasserstein distance metric. Another direction is to extend robust MDPs to a wider family of MDPs, such as the MDP with infinitely many states and with function approximation.

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
