# OpenReview forum: "Online Policy Optimization for Robust MDP"
_ICLR.cc/2023/Conference — Submitted to ICLR 2023_

### Official Review · Reviewer_ZEUJ · 2022-10-24

**Confidence:** 5
**Clarity, Quality, Novelty And Reproducibility:** Please see my major and minor comment…
**Correctness:** 3
**Technical Novelty And Significance:** 2
**Empirical Novelty And Significance:** 2
**Recommendation:** 3

**Strength And Weaknesses:**

1. Most importantly, the contribution of this paper is confusing. While this paper considers an “online” setting without assuming knowledge of the environment, the proposed algorithm requires the size of the ambiguity set as an input. In this spirit, it is not clear to me how the proposed algorithm differs from existing algorithms, such as [i] and [ii].

2. Moreover, the paper assumes that the agent interacts with an unknown nominal environment, but the regret is defined to be the difference between policy pi_k and the optimal robust value function, instead of the optimal value function under the nominal environment.

3. The writing of the technical content is confusing. For example, it seems that the authors attempt the compute the support function sigma(V) for the s-rectangular case; however, for s-rectangular robust MDPs, the corresponding Bellman equation is not the same as the (s,a)-rectangular case, and it does not use the support function.

More comments/questions:

Section 1, 4th paragraph, for (s,a)-rectangular robust MDPs, they have optimal deterministic policies; thus, the argument is not clear.

Section 3, should {P}_{h=1}^H be {P_h}_{h=1}^H?

Section 3, what is P_h = { P_h^o }? Or should it be P_h = P_h^o?

Section 3, before definition 3.2. I don’t think s-rectangularity provides a “stronger robustness guarantee”

Section 3, assuming P_h^o(.|s,a) > 0 appears to be a strong assumption. For example, this assumption does not hold for the classical machine replacement example. Could it be relaxed by perturbing P_h^o? What is the error bound?

Section 3, Learning protocols and regret. What is V_1? Should it be V?

Claim 4.1. This is confusing as this paper assumes that the agent interacts with the nominal environment.

Equation (3). What are the upper and lower bounds when using the bisection method?

Equation (3). Efficiently computing the support function with the L1 ambiguity set is not new. See [i].

Equation (4). To the best of my knowledge, for the s-rectangular case, the support function is not used in the Bellman equation.

Equation (4). Why it could be solved in O(A)?

[i] Marek Petrik, and Dharmashankar Subramanian, RAAM: The Benefits of Robustness in Approximating Aggregated MDPs in Reinforcement Learning, 2014.
[ii] Yue Wang, and Shaofeng Zou, Policy Gradient Method For Robust Reinforcement Learning, 2022.

**Summary Of The Paper:**

The paper considers robust MDPs and proposes an online optimization policy where the agent interacts with the nominal environment.

**Summary Of The Review:**

1. Most importantly, the contribution of this paper is confusing. While this paper considers an “online” setting without assuming knowledge of the environment, the proposed algorithm requires the size of the ambiguity set as an input. In this spirit, it is not clear to me how the proposed algorithm differs from existing algorithms, such as [i] and [ii].

2. The definition of "regret" is less common.

---

> ### Author Response · Authors · 2022-11-08
> **Response to Reviewer ZEUJ (part 1)**
>
> We thank the reviewer for the comments and constructive suggestions. We clarify the concerns  in the following sections. We have also fixed the typos pointed out by the reviewer in our updated manuscript (highlighted in blue color).
>
> We want to emphasize that our main contribution is on the first regret bound for online robust MDP, rather than focusing on the computation complexity. We believe that the term ``online" has confused the reviewer. We refer to the concept in the learning community where the agent updates the policy gradually using limited (bandit-style) feedback from the environment. In our case, in particular, the transition kernel and reward function are unknown initially. We provide a more detailed description for online learning later.
>
> We also wish to point out our algorithm updates the policy with an online mirror descent style algorithm instead of computing the Bellman operator directly under $s$-rectangular set.
>
> # 1. Online learning with robust MDP
> We study the finite-horizon Robust MDP with online interactions. We say that our interactions are "online" as the learner will only receive the rewards corresponding to the states visited. This is often referred to as bandits feedback and hence arises the classic exploration-exploitation dilemma. We refer to [1] (one of the earlier regret guarantees for online reinforcement learning) and its related work on more details of the setting. We also note that this definition of ``online'' is not related to the knowledge of the size of the uncertainty set $\rho$.
>
> Our objective is to learn the best-performing policy under the worst-case transition within an uncertainty set and our performance metric is regret. From the definition of robust MDP and the definition of best-performing policy under which, it is not reasonable to compare the robust value function of the learned policy with the optimal value function under the nominal environment. With this, we define our regret as the cumulative difference between the robust value function of the learned policy and the optimal robust value function of the robust MDP. The assumption of interacting with the nominal transition kernel is also reasonable as without it is NP-hard to obtain low regret (proved in [2]). We remark that this comparison is also used in robust MDP (see [3] and [4] for example).
>
> # 2. Comparison with previous works
> The online setting also distinguishes our work from most of the previous analyses of the robust MDP and related algorithms (including [5] and [6]) which are based on either offline dataset (where samples from the robust MDP are given) or with access to a generative model (is able to query any state-action-state-reward tuple).
>
> Compared to previous work, [5] assumes that data samples from the robust MDP are already given. Beyond that, the learning error bound is also not a finite-sample bound. Similarly, the robust policy gradient method proposed in [6] is under infinite horizon MDP with exact knowledge of the value functions and the visitation distributions. The actor-critic algorithm also requires sampling a trajectory of data at each time step (which would require a simulator to do so). Beyond that, the algorithms only enjoy a finite-sample guarantee for a smoothed objective, which is only equivalent to the original objective asymptotically.
>  similar usages.
>
> ## Reference
> [1] Auer, Peter, Thomas Jaksch, and Ronald Ortner. "Near-optimal regret bounds for reinforcement learning." Advances in neural information processing systems 21 (2008).
>
> [2] Eyal Even-Dar, Sham M Kakade, and Yishay Mansour. Experts in a Markov decision process.
> Advances in Neural Information Processing Systems, 2004.
>
> [3] Panaganti, Kishan, and Dileep Kalathil. "Sample Complexity of Robust Reinforcement Learning with a Generative Model." International Conference on Artificial Intelligence and Statistics. PMLR, 2022.
>
> [4] Yang W, Zhang L, Zhang Z. Towards theoretical understandings of robust markov decision processes: Sample complexity and asymptotics[J]. arXiv preprint arXiv:2105.03863, 2021.
>
> [5] Petrik, Marek, and Dharmashankar Subramanian. "RAAM: The benefits of robustness in approximating aggregated MDPs in reinforcement learning." Advances in Neural Information Processing Systems 27 (2014).
>
> [6] Wang, Yue, and Shaofeng Zou. "Policy Gradient Method For Robust Reinforcement Learning." International Conference on Machine Learning. PMLR, 2022.
>
> [7] Jin, Chi, et al. "Is Q-learning provably efficient?." Advances in neural information processing systems 31 (2018).
>
> [8] Ho, Chin Pang, Marek Petrik, and Wolfram Wiesemann. "Fast Bellman updates for robust MDPs." International Conference on Machine Learning. PMLR, 2018.
>
> [9] Wiesemann, Wolfram, Daniel Kuhn, and Berç Rustem. "Robust Markov decision processes." Mathematics of Operations Research 38.1 (2013): 153-183.

---

> ### Author Response · Authors · 2022-11-08
> **Response to Reviewer ZEUJ (part 2)**
>
> # 3. Definition of regret
> The regret is indeed defined with $V_1$ as this denotes the cumulative reward of the episodes (the definition on the top of page 4). We want to note that this is a standard notation used in online episodic finite-horizon MDP, and we refer to [1],[7] and its related works for similar usages.
>
> # 4. On the update rules for $s$-rectangular set
> Under the $s$-rectangular assumption, as the uncertainty set cannot be separated at each $(s,a)$ pair, solving the primal problem of $\sigma_P(V)$ is difficult. Thus a different Bellman operator may be used (e.g. eq.5 of [8]). However, our algorithm is not computing the Bellman operator under $s$-rectangular set. In fact, the update rule of the policy evaluation does not enjoy the basic properties of the Bellman operator, such as contractive property. The optimization of the policy is instead performed in the policy improvement stage, through online mirror descent. Thus, in the policy evaluation step, the policy may be considered to be fixed. Hence we are not solving the whole $\max_\pi \min_P$ problem in the Bellman operator (eq.5 of [8]).
>
> # 5. Clarification on Claim 4.1
> Our Claim 4.1 suggests the importance of considering a robust policy for minimizing the regret in the robust MDP (while interacting with the nominal transition). Specifically, the claim is saying that the optimal policy under the nominal environment can be arbitrarily bad (even worse than a uniformly random policy).
>
> # 6. On the ``stronger robustness guarantee'' of $s$-rectangularity
> On the assumption of $s$-rectangular uncertainty set. We say that the $s$-rectangular uncertainty set may offer a stronger robustness guarantee as it relaxes on the assumption of the adversarial perturbation of the transition function. Under $(s,a)$-rectangular set, the perturbation is limited to those that are independent for each $(s,a)$ pair. In contrast, this is relaxed in the $s$-rectangular set as the perturbations can be only independent for each state $s$. We refer to [9] for a more detailed discussion and comparison between the two uncertainty sets.
>
> # 7. On solving eq.3 and eq.4
> We mentioned bisection algorithms to showcase the feasibility to compute the problems. As our main contribution is not on proposing new ways to compute the inner maximization problem, we believe that the exact computational complexity and error bounds of computing them are out of the scope of this paper. We thank the reviewer for pointing out that eq.4 cannot be exactly solved in $O(A)$ (but can still be solved efficiently with linear programming methods) and we have corrected this in our updated manuscript.
>
> # 8. Clarification on the $4$-th paragraph
> We thank the reviewer for pointing out the confusing description. We have updated paragraph 4 to avoid confusion. We were referring to the fact that the optimal policy on robust MDP may be randomized (under $s$-rectangular set).
>
> # 9. On relaxing the assumption $P_h^o(\cdot \mid s,a) > 0$
> We believe that this assumption may be relaxed through other assumptions and the current assumption may be an artifact of analysis. We note that this assumption is also implicitly used in previous robust MDP analyses, such as [3]. We have pointed this out in the future direction section.
>
> However, we remark that MDPs that do not satisfy this assumption can be slightly perturbed to satisfy this. We believe that adding an arbitrary noise $\epsilon/(SAH)$ to each $P_h^o(s^\prime \mid s,a)$ may suffice. If the rewards are bounded between $[0,1]$, then the robust value function will have a final error bound of $\epsilon$.

---

> ### Author Response · Authors · 2022-12-06
> **Thank you and we welcome further comments.**
>
> We thank the reviewer again for the feedback. We hope that most of the concerns could have been addressed by this discussion. If there are any further questions and comments, on the manuscript, we are very happy to follow up and discuss them.

---

### Official Review · Reviewer_QpYY · 2022-10-25

**Confidence:** 4
**Correctness:** 3
**Technical Novelty And Significance:** 2
**Empirical Novelty And Significance:** 3
**Recommendation:** 6

**Clarity, Quality, Novelty And Reproducibility:**

The clarity and quality is very good.
The reproducibility of the experiments may need some more details such as the parameters of the perturbed transition dynamics.

The novelty of the results is great. The novelty of the technical details is a little bit hard to see. It may be better to give more discussions about the main technical tools used to deal with the challengs.


**Strength And Weaknesses:**

Strength:
1. This is a very interesting topic whether some algorithms can achieve $O(\sqrt{K}$ regret in an online robust RL case with an uncertainty set. The results are of a somewhat significant novelty since this work gives the first provable policy optimization algorithm with a non-asymptotic regret bound and sub-linear sample complexity.
2. I didn't check the proof in the appendix, but the writing and presentation are almost clear and easy to follow.

Weaknesses:
1. This work claim that the algorithm is designed by adaptation to the robust case from the non-robust algorithm in [2]. However, [2] seems not the best non-robust policy-based algorithm for online RL in the literature with regret bound $O(\sqrt{S^2AH^4K})$. Since the optimal policy-based algorithm is [1] with bound $O( \sqrt{SAH^3K}+ \sqrt{AH^4K} )$ is better than [2], it seems to apply a robust version of the algorithm in [1] will lead to a better policy and better results. It is not clear why this is not considered.
2. The sample complexity of prior works in Table 1 seems wrong if I didn't miss something. In $(s,a)$-rectangular case, if the claimed sample complexity means the required number of episodes, the results of [D] is $O(\frac{H^4S^2A(2+\rho)^2}{\rho^2 \epsilon^2})$, and also the $s$-rectangular case.
2. The difficulty of addressing additional challenges due to the combination of robustness requirements and the online setting is not introduced in detail. Even if there is a section describing Challenges in Optimistic Robust Policy Evaluation, it will be better to have more discussions. Or it will seem like a combination of the techniques in non-robust online RL and robust RL.
3. In the experiments, the parameters of the perturbed transition dynamic are not given in detail. So it is a little bit hard to rerun the experiments or check the results.

Some extra questions or minors:
1. Why the sample complexity does not depends on the uncertainty level with uncertainty level $\ell_1$? That means the sample complexity does not depend on it? If so, how to compare the results with [Yang et al. 2021] with the $\rho$ in the denominator of the sample complexity.
2. The first equation on Page 5 may have typos if I didn't miss something. should $s_0$ be $s_0^k$?

[1] Wu, Tianhao, et al. "Nearly optimal policy optimization with stable at any time guarantee." International Conference on Machine Learning. PMLR, 2022.
[2] Shani, Lior, et al. "Optimistic policy optimization with bandit feedback." International Conference on Machine Learning. PMLR, 2020.



**Summary Of The Paper:**

This paper is the first provable policy optimization algorithm for robust MDP in online RL with a $\ell_1$ uncertainty set, along with a finite-sample complexity bound. The main contribution is that compared to the previous sample complexity of robust MDP, under an online RL setting, it gives the first provable optimization algorithm with a non-asymptotic regret bound and sub-linear sample complexity. The sample complexity also recovers a non-robust policy optimization result of a prior work when the uncertainty set $\rho \rightarrow 0$. Experiments are conducted to show the results.

**Summary Of The Review:**

I recommend borderline acceptance since this is the first provable policy-based robust algorithm with finite-sample complexity. However, as the technical tools are not well-introduced, it is hard to see whether this is a combination of the tools in non-robust online RL and robust RL or far beyond. The results are somewhat significant and the technical tools may be novel but it is hard to see owing to no discussions.

---

> ### Author Response · Authors · 2022-11-08
> **Response to Reviewer QpYY**
>
> We thank the reviewer for the insightful comments and constructive suggestions. We address and clarify the major concerns raised in the following sections.
>
> # 1. Potential use of the stable-at-any-time technique
> We agree with the reviewer that, with the techniques proposed in [1], one can modify our algorithm to recover the near-optimal regret for non-robust policy optimization when $\rho = 0$ and $S > H$. As we focus more on the consideration of robustness, we did not employ their techniques for simpler analysis.
>
> # 2. Details of the experiments
> Reproducibility of the experiments: Our experiments were conducted with the GridWorld environment provided by rlberry with additional perturbation to the transition. The perturbation is chosen in the opposite direction of the agent's direction. We will make our experiment codes and our environments open-sourced to ensure the reproducibility of our results.
>
> # 3. Dependency on $\rho$
> We first would like to remark on the fact that the sample complexity under $\ell_1$ distance set independent of $\rho$ is not entirely new. For example, Theorem 1 of [1] also showed that the sample complexity can be independent of $\rho$ (which is the same as ours). One possible explanation of this is because, by the definition of $\ell_1$ distance, $\rho$ is upper bounded by $2$ (which is a constant, independent of the state, and action size).
>
> In comparison to results in [2], Theorem 3.1 and 3.2 of [2] show that the sample complexity is $\tilde{O}\left((2+\rho)^2 / \rho^2 \right) \geq 1$. Thus the result will be closer to the non-robust result when $\rho$ is not small and will explode when $\rho$ is 0. When $\rho$ is very large, [2]'s result is still strictly worse than ours in terms of $\rho$. However, when $\rho$ is small, our result will not explode and can recover the result for non-robust policy optimization when $\rho = 0$. We also want to remark that the theoretical results provided [2] (Theorem 3.1 and 3.2) are only for policy evaluation.
>
> # 4. Remark on proof techniques
> We thank the reviewer for the suggestion and we have updated the discussion of proof sktech in our main paper.
> To summarize, the highlights of our analysis is on (1) the common value difference lemma in policy optimization algorithm analysis no longer holds and (2) the existing robust MDP analysis does not imply a valid optimism bonus.
> We also highlight the key differences between our work and non-robust policy optimization as follows.
>
> **Common value difference lemma in non-robust policy optimization cannot be applied.**
>
> In contrast to non-robust policy optimization, the robust policy optimization analysis is highly dependent on the varying robust transitions. An immediate consequence of this is that the common value difference lemma (Lemma 1 of [4], Lemma 4.2 of [5]) can no longer be directly applied. Naively employing a recursive relation with respect to a fixed transition kernel in a similar way to the value difference lemma may lead to linear regret. Thus, we instead perform a recursion conditioned on varying transition kernels. In this case, maintaining optimism is hard as the expectation of each time step $h$ is taken with respect to a different transition kernel.
>
> **Addressing multiple uncertainties simultaneously in robust MDP.**
>
> In contrast to non-robust policy optimization, the learner now has to tackle this uncertainty and the uncertainty from the robust MDP simultaneously. To establish an optimism bonus for the uncertainty of the transition caused by limited interaction and the uncertainty set, we derive the dual formulation of inner optimization problem $\sigma_{\hat{\mathcal{P}}_{(s,a)}}(V)$. This allows us to decouple the uncertainty and bound each source of uncertainty separately. We then show that the dual variable $\eta$ must be bounded at its optimum by inspecting certain pivot points and by the convexity of the dual. When we have such bounds of $\eta$, applying Hoeffding's type concentration over it with an $\epsilon$-net argument will yield the desired regret bound.
>
> # 5. Response to other comments
> * We thank the reviewer for pointing out the typo in Table 1. The sample complexity result in [D] is now corrected.
> * We have corrected the typo of the first equation on Page 5 in our updated manuscript.
>
> ## Reference
> [1] Panaganti, Kishan, and Dileep Kalathil. "Sample Complexity of Robust Reinforcement Learning with a Generative Model." International Conference on Artificial Intelligence and Statistics. PMLR, 2022.
>
> [2] Yang W, Zhang L, Zhang Z. Towards theoretical understandings of robust markov decision processes: Sample complexity and asymptotics[J]. arXiv preprint arXiv:2105.03863, 2021.

---

### Official Review · Reviewer_JsyR · 2022-10-25

**Confidence:** 5
**Clarity, Quality, Novelty And Reproducibility:** In Strength And Weaknesses
**Correctness:** 3
**Technical Novelty And Significance:** 2
**Empirical Novelty And Significance:** Not applicable
**Recommendation:** 5

**Strength And Weaknesses:**

This paper extends robust MDP with a generative model/ offline dataset to the online setting by optimizing policies. This is the first online algorithm in robust MDPs. Though the extension is mainly based on prior works for non-robust MDPs, which means this paper's novelty is limited, I think this work is fundamental. I didn't go through all the details and appendix in this paper due to my heavy review load. Though I want to give the score 4, it is not in the choice. The authors should know I can't be more positive at the current stage by giving score 5. I will leave the decision to AC.

Below I list some questions and comments that I hope the authors could answer.

1. As we know, in $L_1$ uncertainty set, the size $\rho$ has an upper bound $2$. Thus, the result in Theorem 1 has nothing to do with $\rho$. Then, what's the role of robustness? By [1], we know that when $\rho$ is not small, the sample complexity could be reduced. But I can't see such similar results from this paper.
2. About Eqn (4), why it is different from the expression in [1] (Lemma B.6), which means Eqn (4) is independent with policy $\pi$?
3. I would encourage the authors to pay more attention to their proof sketch. For example, they could illustrate how they do error decomposition and how to control each error term. The current description in Sec 5 is more like a redundancy.
4. A missing related work. I believe the authors should discuss with it.[2]


[1] Yang W, Zhang L, Zhang Z. Towards theoretical understandings of robust markov decision processes: Sample complexity and asymptotics[J]. arXiv preprint arXiv:2105.03863, 2021.

[2] Liu Z, Bai Q, Blanchet J, et al. Distributionally Robust $ Q $-Learning[C]//International Conference on Machine Learning. PMLR, 2022: 13623-13643.

**Summary Of The Paper:**

In Strength And Weaknesses

**Summary Of The Review:**

In Strength And Weaknesses

---

> ### Author Response · Authors · 2022-11-08
> **Response to Reviewer JsyR**
>
> We thank the reviewer for the very insightful comments. However, we respectfully disagree that extending the existing optimistic algorithm is of limited novelty. In particular, we highlight two items in our response.
>
> # Remark of our results and analysis
> First, we would like to remark that the proof of lemma B.6 in [2] may have some mistakes and thus their expression may be incorrect.
> Their objective optimization function (Line 9 page 49) is inconsistent with the equality of function in the robust Bellman operator $\Gamma_{r}^\pi V(s)$. We would also like to note that this could affect their subsequent analysis (beyond Lemma B.6) and thus cannot be used to derive regret bound. Our Eq.4 is a correction of this.
>
> Second, following the reviewer's suggestion, we highlight the key differences between our work and non-robust policy optimization. To summarize, the main difference is on (1) the standard value difference lemma no longer holds and (2) the existing robust MDP analysis does not imply a valid optimism bonus. The details are as follows.
>
> **(1) Standard value difference lemma in non-robust policy optimization cannot be applied.**
>  In contrast to non-robust policy optimization, the robust policy optimization analysis is highly dependent on the varying robust transitions. An immediate consequence of this is that the common value difference lemma (Lemma 1 of [4], Lemma 4.2 of [5]) can no longer be directly applied. Naively employing a recursive relation with respect to a fixed transition kernel in a similar way to the value difference lemma may lead to linear regret. Thus, we instead perform a recursion conditioned on varying transition kernels. In this case, maintaining optimism is hard as the expectation of each time step $h$ is taken with respect to a different transition kernel.
>
> **(2) Addressing multiple uncertainties simultaneously in robust MDP.**
>
>  In contrast to non-robust policy optimization, the learner now has to tackle this uncertainty and the uncertainty from the robust MDP simultaneously. To establish an optimism bonus for the uncertainty of the transition caused by limited interaction and the uncertainty set, we derive the dual formulation of inner optimization problem $\sigma_{\hat{\mathcal{P}}_{(s,a)}}(V)$. This allows us to decouple the uncertainty and bound each source of uncertainty separately. We then show that the dual variable $\eta$ must be bounded at its optimum by inspecting certain pivot points and by the convexity of the dual. When we have such bounds of $\eta$, applying Hoeffding's type concentration over it with an $\epsilon$-net argument will yield the desired regret bound.
>
> The above edits are included in the updated draft.
>
> # 1. Form of Eq.4
>
> We thank the reviewer for pointing out the confusion. Eq.4 is the inner optimization problem involved when computing the Q-value function, which takes a $(s,a)$ pair.  In this case, the $a$ is fixed. We remark that [2]'s expression of the inner problem is for computing the robust value function, which is different from ours.
>
> # 2. Dependency on $\rho$
>
> Theorem 3.1 and 3.2 of [2] show that the sample complexity is $\tilde{O}\left((2+\rho)^2 / \rho^2 \right) \geq 1$. Thus when $\rho$ is very large, [2]'s result is strictly worse than ours in terms of $\rho$. When $\rho$ is small, their result tends to infinity while our result will not explode and is able to recover the result for non-robust policy optimization when $\rho = 0$.
>
> The regret upper bound is indeed independent of $\rho$, asymptotically. This is intuitive though. When the algorithm subtly characterizes the uncertainty from all sources, it derives a robust enough policy in a way that if there is a policy that achieves a high return then this policy achieves a high return. The difference between this remains sublinear for any $\rho$. Meanwhile, notice that Theorem 1 of [1] also shows that the sample complexity bound can be independent of $\rho$ under $(s,a)$-rectangular set and access to simulators.
>
> #3. Missing related work
> We thank the reviewer for pointing out the missing related work [3] and we have included a short discussion of it in our updated manuscript. The key difference between their work and ours is in the assumption of a simulator. In contrast, our work is in the online setting where data is acquired sequentially. Without access to a simulator, we remark that the learner faces the well-known dilemma of exploration-exploitation with online interactions. In addition, their objective is to only solve for the asymptotically optimal Q-function, while we focus on deriving an efficient algorithm with sublinear regret.

---

> > ### Comment · Reviewer_JsyR · 2022-11-08
> > **Response and Further Questions**
> >
> > I would like to thank the authors' detailed responses. However, according to the authors' responses, I have more questions about this paper and still cannot be more positive at the current stage.
> >
> > 1. The authors argue that Lemma B.6 in [2] may be incorrect. However, I referred to the original robust bellman equation in [1], it satisfies $\Gamma_r^\pi V (s) = R^\pi(s)+\gamma\inf_{P_{s}\in\mathcal{P}_{s}} \sum^{s',a} P(s'|s, a)\pi(a|s) V(s')$. And I have checked the proof of Lemma B.6 and found no errors. I would appreciate it if the authors could point out the exact error of Lemma B.6 in [2].
> >
> > 2. About Eq.(4). I referred to the authors' appendix and just found that the author takes $\inf_{P\in\mathcal{P}} P(\cdot|s,a) V$ in $s$-rectangular setting. But I also referred to [3,4], and I found it should be $\inf_{P\in\mathcal{P}}\sum_{a}\pi(a|s)P(\cdot|s,a)V$ in $s$-rectangular setting. Reaching this point I can see why the dual objective Eq.(4) is inconsistent with Lemma B.6 in [2], where they chose the form of [3,4]. I believe the authors should explain whether their new definition in $s$-rectangular setting is reasonable.
> >
> > 3. Though the upper bound in [2] didn't tell us the whole story. I think the discussion part in Sec 3.3 in [2] and lower bound in Corollary 3.1 tells us what's the role of robustness. In authors' manuscript, the regret reflects the robustness is useless when $\rho$ is small, which is consistent with [2]'s discussion. However, the regret will be extremely large when $\rho$ is approximately large while [2]'s result is upper bounded. I wish the authors could explain more about their results.
> >
> > 4. I'm also wondering why the results of [2] in Table 1 are only for policy evaluation.
> >
> > [1] G. Iyengar. Robust dynamic programming. Math. Oper. Res., 30:257–280, 05 2005.
> >
> > [2] Yang W, Zhang L, Zhang Z. Towards theoretical understandings of robust markov decision processes: Sample complexity and asymptotics[J]. arXiv preprint arXiv:2105.03863, 2021.
> >
> > [3] Chin Pang Ho, Marek Petrik, and Wolfram Wiesemann. Fast bellman updates for robust
> > mdps. In Proceedings of the 35th International Conference on Machine Learning, pages
> > 1979–1988, 2018.
> >
> > [4] Chin Pang Ho, Marek Petrik, and Wolfram Wiesemann. Partial policy iteration for
> > l1-robust markov decision processes. Journal of Machine Learning Research, 22:275,
> > 2021.

---

> > > ### Author Response · Authors · 2022-11-09
> > > **Further response to Reviewer JsyR's questions**
> > >
> > > We thank the reviewer for the very insightful comments. We respond to the further questions raised in the following section:
> > >
> > > # 1. Lemma B.6 of Yang et al.
> > >
> > > We agree that the robust bellman equation
> > >
> > > $$
> > > \Gamma_r^\pi V(s) =R^\pi(s)+\gamma \inf_{P_s \in \mathcal{P}_s} \sum_\{s^{\prime} \in \mathcal{S}, a \in \mathcal{A}} P\left(s^{\prime} \mid s, a\right) \pi(a \mid s) V\left(s^{\prime}\right)
> > > $$
> > > in [1] is correct.
> > > However, their inner optimization in line 9 of page 49 is
> > > $$
> > > \inf_P \sum_\{s^\prime \in \mathcal{S}} P_a(s^\prime \mid s,a) \pi(a \mid s) V(s^\prime).
> > > $$
> > > Note that this is different from the inner optimization problem in the robust Bellman equation,
> > > $$
> > > \inf_\{P_s \in \mathcal{P}_s} \sum_\{s^{\prime} \in \mathcal{S}, a \in \mathcal{A}} P\left(s^{\prime} \mid s, a\right) \pi(a \mid s) V\left(s^{\prime}\right)\,.
> > > $$
> > > It seems to us that they might have ignored the summation over action $a$ when solving the inner optimization problem, which then may affect their subsequent analysis.
> > >
> > > # 2. Our inner optimization problem for $s$-rectangular set
> > > We apologize that $\sigma_\{\hat{\mathcal{P}}_\{h}}(\hat{V}_\{h+1}^\{\pi})(s)$ is a typo and should be corrected to $\sigma_\{\hat{\mathcal{P}}_\{h}}(\hat{V}_\{h+1}^\{\pi})(s,a)$. We note that $\sigma_\{\hat{\mathcal{P}}_h}(\hat{V}_\{h+1}^\{\pi})(s,a)$ is defined in eq.1 while the typos are made in the appendix. We have updated this in our manuscript and realized that this might have misled the reviewer. The main cause of the difference between our results and [1] is that they computed the robust value while we computed the robust $Q$-value function (which takes a $(s,a)$-pair)
> > > $$
> > > \hat{Q}^\{k}_\{h} (s,a) = \min\left(\hat{r}(s,a) + \sigma_\{\hat{\mathcal{P}}_\{h}}(\hat{V}_\{h+1}^\{\pi})(s,a) + b_h^k(s,a), H\right).
> > > $$
> > > Therefore, our inner optimization problem is with a fixed action $a$,
> > > $$
> > > \sigma_\{\hat{\mathcal{P}}_\{h}}(\hat{V}_\{h+1}^\{\pi})(s,a) = \inf_\{P_s \in \mathcal{P}_s} \sum_\{s^{\prime}} P\left(s^\prime \mid s, a\right)  V\left(s^{\prime}\right).
> > > $$
> > >
> > > # 3. Bounded regret when $\rho$ is large
> > > We would first like to note that our regret would not be arbitrarily large in the large $\rho$ regime. This is due to the definition of $\ell_1$ distance, which implies $0 \leq \rho \leq 2$. We also note that the Corollary 3.1 of [1] is not a "general" lower bound, in the sense that they have restricted $\rho = (2\gamma - 1)/ \gamma$, where $\gamma$ is the discount factor in the infinite horizon MDP. In the case of the finite horizon, we set $\gamma = 1$ and their choice of $\rho$ is then $\rho = 1$. In this case, their sample complexity result is $\Omega \left( (SA (1 - \gamma)) / (\epsilon^2(1 - \gamma)^4\right)$, which is analogous to our result.
> > >
> > > # 4. Clarification on the "policy evaluation"
> > > The theoretical results in Theorem 3.1 Theorem 3.2, [1], show a sample complexity bound of $V^\pi - \hat{V}^\pi$ (uniformly over all $\pi$), where $V^\pi$ is the robust value function, $\hat{V}^\pi$ is the approximated robust value function obtained through updates according to the robust Bellman equation. We regard the above-mentioned mapping from $\pi$ to $\hat{V}^\pi$ as \textit{policy evaluation}. The sample complexity bound they provide is also for this uniform estimation only.
> > >
> > > However, [1] did not further discuss how one could find the optimal value function and the optimal policy: Maximizing over all policies $\hat{\pi}^\ast = \arg\max_{\pi} \hat{V}^\pi$ may not be feasible as the optimal robust policy could be stochastic (thus the policy space may be infinite); Although not being discussed by [1], under access to a generative model, one could further run policy iteration methods with the estimated value function. Yet, this additional step is not included in the sample complexity bound.
> > >
> > > We realized that calling [1] as *policy evaluation* can be confusing and misleading, hence we have removed the "PE" note from Table 1 when discussing [1]'s results.
> > >
> > > ## Reference
> > > [1] Yang W, Zhang L, Zhang Z. Towards theoretical understandings of robust Markov decision processes: Sample complexity and asymptotics [J]. arXiv preprint arXiv:2105.03863, 2021.
> > >
> > > We hope that this clarifies the reviewer's concerns and we welcome further questions.

---

> > > > ### Comment · Reviewer_JsyR · 2022-11-09
> > > > **Responses**
> > > >
> > > > Thanks for the authors' responses.
> > > >
> > > > 1. I just re-read [2]'s Line 9 in Lemma B.6. But I think it's a typo where the authors' of [2] forgot to add it. As I have derived the duality of Lemma B.6 with the correct primal problem, and my results match with the final results of Lemma B.6. But I'm still wondering why this might affect their subsequent analysis.
> > > >
> > > > 2. I think the authors may misunderstand my point here. As I referred to [2,3,4], under s-rectangular assumption, the robust bellman operator is defined by
> > > >
> > > >     $\inf_{P_s\in\mathcal{P}_s}\Sigma_a \pi(a|s)P(\cdot|s,a) V$.
> > > >
> > > >     However, in this paper, the authors use $\inf_{P_s\in\mathcal{P}_s}P(\cdot|s,a) V$. As far as I know, there are no theories with this definition. I would appreciate it if the authors could explain the reasonability of their different definition under s-rectangular assumption.
> > > >
> > > > 3. Yes I just realized the range of $\rho$ is upper bounded by 2. However, I believe Corollary 3.1 in [2] didn't set $\rho=(2\gamma-1)/\gamma$ but indeed $p=(2\gamma-1)/\gamma$, which is a fundamental technique from [5] I think. Thus I'm quite confused by the authors' response to this point. Besides, my original point here is Corollary 3.1 tells us $\rho$ might be useless when it is small. But when $\rho>1-\gamma$, robustness plays a role to reduce the sample complexity. However, compared with this paper's result, I didn't see a similar phenomenon. It seems the regret is increasing w.r.t. $0\le\rho\le2$. I would appreciate it if the authors could explain why robustness will enlarge the regret.
> > > >
> > > > 4. That makes sense. For PE, according to [2], the sample complexity is smaller with a fixed policy $\pi$ in s-rectangular setting, because the covering number of policy space is larger than that of value space and PE can skip the union bound over policy space. However, I also checked the empirical results of [2] indeed. I find they have applied the modified Bisection algorithm [6] to obtain the optimal robust value function in s-rectangular assumption. But I didn't find how the optimal robust policy is obtained in [2]. It seems they indeed have calculated the optimal robust policy to construct their confidence interval. I guess they applied Theorem 4 in [6] to calculate it.
> > > >
> > > > [1] G. Iyengar. Robust dynamic programming. Math. Oper. Res., 30:257–280, 05 2005.
> > > >
> > > > [2] Yang W, Zhang L, Zhang Z. Towards theoretical understandings of robust markov decision processes: Sample complexity and asymptotics[J]. arXiv preprint arXiv:2105.03863, 2021.
> > > >
> > > > [3] Chin Pang Ho, Marek Petrik, and Wolfram Wiesemann. Fast bellman updates for robust mdps. In Proceedings of the 35th International Conference on Machine Learning, pages 1979–1988, 2018.
> > > >
> > > > [4] Chin Pang Ho, Marek Petrik, and Wolfram Wiesemann. Partial policy iteration for l1-robust markov decision processes. Journal of Machine Learning Research, 22:275, 2021.
> > > >
> > > > [5] Mohammad Gheshlaghi Azar, R´emi Munos, and Hilbert J Kappen. Minimax pac bounds
> > > > on the sample complexity of reinforcement learning with a generative model. Machine
> > > > learning, 91(3):325–349, 2013.
> > > >
> > > > [6] Chin Pang Ho, Marek Petrik, and Wolfram Wiesemann. Fast bellman updates for robust
> > > > mdps. In Proceedings of the 35th International Conference on Machine Learning, pages
> > > > 1979–1988, 2018.

---

> > > > > ### Author Response · Authors · 2022-11-09
> > > > > **Response to additional questions raised by Reviewer JsyR**
> > > > >
> > > > > # 1. On Lemma B.6 of [2] and on the robust Bellman operator
> > > > > We note that we have used the inner optimization problem to update the $Q$-value function (which is not by the robust bellman equation $V = \Gamma V $)
> > > > > $$\hat{Q}^\{k}_\{h} (s,a) = \min\left(\hat{r}(s,a) + \sigma_\{\hat{\mathcal{P}}_\{h}}(\hat{V}_\{h+1}^\{\pi})(s,a) + b_\{h}^\{k}(s,a), H\right) .$$
> > > > > Since we are not using the robust bellman equation $V = \Gamma V$, and we are computing the $Q$-value function, which takes a $(s,a)$-pair as input, our inner problem is thus different and thus not involve the $\pi$ term explicitly.
> > > > > This distinguishes our update rule and our inner optimization problem from [2,3,4]. This is also the **main** reason why our dual optimization problem is different from [2]. We argue that updating the $Q$-value function with a fixed $(s,a)$-pair may be favored is that the robust bellman operator is known to be hard to compute efficiently under $s$-rectangular set due to the coupling with $\pi$.
> > > > >
> > > > > We thank the reviewer for pointing out that the summation in Lemma B.6 is a typo and can be fixed. However, we still find the subsequent analysis following Lemma B.6 inapplicable to our result as there are other inconsistent steps (but can be fixed by our analysis).
> > > > > For example, on page 50, [2] constrain the dual variable to be $\sum_\{a} \eta_\{a} \leq \frac{2 + \rho}{ \rho ( 1 - \gamma)}$. This relies on an earlier step $
> > > > > \max_\{a}\left(\frac{\eta_\{a}-\pi(a) V_\{\min}}{2}\right)_{+} \geq \frac{1}{|\mathcal{A}|} \sum_\{a}\left(\frac{\eta_\{a}-\pi(a) V_\{\min}}{2}\right)_\{+}
> > > > > $, which, if not mistaken, may also be of concern. Our proof instead argues the optimality of each dual variable $\eta_a $ individually, which leads to a different result.
> > > > >
> > > > > # 2. On the lower bound
> > > > > We thank the reviewer for pointing out that we have confused $p$ with $\rho$. We would like to explain the difference between the two results from two points.
> > > > >
> > > > > (1) The lower bound provided by [2] is for policy evaluation and shows the minimum sample needed to estimate robust value function for any policy $\pi$. Our regret bound instead implicitly provides a guarantee for the amount of online interaction needed to learn the optimal robust value function and the optimal robust policy. We remark that as our regret compares with the optimal robust policy, the lower bound provided by [2] can be strictly greater than the lower bound in our case. Thus we believe that it is hard to infer the correct dependency on $\rho$ in regret from [2]'s result.
> > > > >
> > > > > (2) Our work investigates the online setting, which requires a policy to interact (explore and exploit) the environment to collect samples. Thus when $\rho$ is large, this implies that our policy needs to consider a wide range of environmental models. Under this case, exploration and exploitation may be harder to balance than in the non-robust case (when $\rho = 0$). From this perspective, it is reasonable to expect the regret to be increasing when $\rho$ is large.
> > > > >
> > > > > # 3. On "policy evaluation"
> > > > > We agree with the reviewer that such empirically robust policy may be derived when value functions are obtained through methods provided in [2]. We would like to point out that, though this is empirically feasible, [2] did not provide any theoretical guarantees for it.
> > > > >
> > > > > We hope that these will clarify the reviewer's concerns and we welcome any further questions.
> > > > >
> > > > > ## Reference
> > > > > [1] G. Iyengar. Robust dynamic programming. Math. Oper. Res., 30:257–280, 05 2005.
> > > > >
> > > > > [2] Yang W, Zhang L, Zhang Z. Towards theoretical understandings of robust markov decision processes: Sample complexity and asymptotics[J]. arXiv preprint arXiv:2105.03863, 2021.
> > > > >
> > > > > [3] Chin Pang Ho, Marek Petrik, and Wolfram Wiesemann. Fast bellman updates for robust mdps. In Proceedings of the 35th International Conference on Machine Learning, pages 1979–1988, 2018.
> > > > >
> > > > > [4] Chin Pang Ho, Marek Petrik, and Wolfram Wiesemann. Partial policy iteration for l1-robust markov decision processes. Journal of Machine Learning Research, 22:275, 2021.

---

> > > > > > ### Author Response · Authors · 2022-11-16
> > > > > > **Thank you and we welcome further questions and comments.**
> > > > > >
> > > > > > We thank the reviewer again for the constructive feedback. We very much enjoy the discussion with the reviewer and this helps us to improve the work and the presentation. We hope that most of the concerns could have been addressed by this discussion. If there are any further questions and comments, on the manuscript, we are very happy to follow up and discuss them.

---

> ### Author Response · Authors · 2022-11-08
> **Reference to the response to Reviewer JsyR**
>
> ## Reference
> [1] Panaganti, Kishan, and Dileep Kalathil. "Sample Complexity of Robust Reinforcement Learning with a Generative Model." International Conference on Artificial Intelligence and Statistics. PMLR, 2022.
>
> [2] Yang W, Zhang L, Zhang Z. Towards theoretical understandings of robust markov decision processes: Sample complexity and asymptotics[J]. arXiv preprint arXiv:2105.03863, 2021.
>
>
> [3] Liu Z, Bai Q, Blanchet J, et al. Distributionally Robust Q-Learning[C]//International Conference on Machine Learning. PMLR, 2022: 13623-13643.
>
> [4] Shani, Lior, et al. "Optimistic policy optimization with bandit feedback." International Conference on Machine Learning. PMLR, 2020.
>
> [5] Cai, Qi, et al. "Provably efficient exploration in policy optimization." International Conference on Machine Learning. PMLR, 2020.

---

### Official Review · Reviewer_yGXw · 2022-11-03

**Confidence:** 4
**Correctness:** 3
**Technical Novelty And Significance:** 3
**Empirical Novelty And Significance:** 3
**Recommendation:** 6

**Clarity, Quality, Novelty And Reproducibility:**

This paper is well-written and easy to follow in most places. For reproducibility, the details of the environment are provided. The solid theoretical result is novel and is expected to have good impacts to the RL community.

**Strength And Weaknesses:**

**Strength**

- A solid paper supported by theoretical regret analysis.
- This is the first work that provides an algorithm with theoretical regret bounds under robust MDP in the online setting.
- Proposition 4.1 that shows the sub-optimality of the policy learned from the nominal transition model is interesting and provides a good motivation for ROPO.
- The paper is well-written and easy to follow.

**Weaknesses**
- There are some issues regarding the main technical analysis:
    - The statement “previous sample complexity results cannot directly imply a sublinear regret” does not make much sense to me as it seems feasible to convert the sample complexity results in Table 1 to regret bounds. Why cannot the regret bound of [A] in Table 1 be converted from the sample complexity result?
    - In the proof of Lemma 4, “L(\tilde{\eta}, \lambda)(s, a) is inversely proportional to \lambda” is not true since \tilde{\eta} also depends on \lambda. As a result, the optimal \lambda derived in P.16 appears incorrect, and this could affect the subsequent analysis.
Should Eq.3 be a “constrained” optimization problem? Specifically, in the proof of Lemma 4, it is required that $\tilde{\eta} - min_s \hat{V}(s) \leq \lambda$. And the issue also arises in Eq.4. It is not immediately clear whether this would affect the analysis.
    - In the second paragraph of P.3, the description about Badrinath & Kalathil (2021) is incorrect. It appears that the theoretical result of this work only provides asymptotic guarantees instead of a convergence rate.
- The experiment results only show the performance of ROPO under the (s, a)-rectangular sets, lacking the result under the s-rectangular uncertainty sets.

Additional comments:

1. Does the policy improvement step in Algorithm 1 require $-$?
2. In the proof of Theorem 1, “ By Lemma 2 and Lemma 4,” => “ By Lemma 2 and Lemma 3,”
3. The simplex set of the uncertainty set shall be $\Delta_{S*A*H}$?
4. What does it mean to have "a fixed step of time-dependent uncertainty kernels" mentioned at the beginning of Section 3?
5. The definition of $\Delta_S$ in Section 3 is not provided.
6. The statement “This characterization is then more general, and its solution gives a stronger robustness guarantee.” on top of Definition 3.2 does not make much sense.
7. Is it still NP-hard to solve the robust MDP under the two assumptions of the transition kernel?
8. There seems to be a typo that the estimator of the transition is conditioned on $s’$.
9. $\hat{V}$ is not defined.
10. The value function used in Eq.4 should be the estimator of value function \hat{V}?
11. The statement above the “Policy Improvement Step" is weird. \rho and $\hat{P}$ seem to be in different terms.
12. For the step of policy improvement, should “argmin” be “argmax”?
13. How to define the gradient of \hat{V}^{\pi_k}?



**Summary Of The Paper:**

This paper studies the online robust MDP and proposes a policy optimization algorithm called Robust Optimistic Policy Optimization (ROPO) for achieving sublinear regrets. This paper starts by showing a sub-optimality result of non-robust optimal policies, which motivates the study of ROPO. The authors then provide two kinds of regret bounds of ROPO under different assumptions of the underlying uncertainty set of transition kernels. The result appears to be the first regret bound for policy-based methods in the online robust RL setting. Finally, this paper presents experimental results in simple RL environments to corroborate the performance.

**Summary Of The Review:**

This paper takes the first step towards establishing the theoretical regret bounds for the online robust MDP. The contribution is significant as the technique introduced in this paper is valuable for sparking further results on online robust RL. My current rating is 6 mainly due to the concerns mentioned above. The score would be reconsidered if these issues can be addressed during the rebuttal.

---

> ### Author Response · Authors · 2022-11-08
> **Response to Reviewer yGXw**
>
> We thank the reviewer for the very helpful comments. We address the major concerns in the following sections. In addition, we have also addressed the typos pointed out by the reviewer in our updated manuscript (highlighted in blue).
>
> # 1. Conversion of [A]'s sample complexity result to regret bound
> In general, sample complexity results cannot be directly converted into a regret bound (see [2]) for a more detailed discussion. Moreover, [A]'s the result is obtained under access to a generative model, and thus cannot be applied to the online setting. Therefore we did not convert the result to a regret bound in our manuscript. However, by taking a specific value of $K$, we can convert [A]'s result to a loose regret bound of $O\left(
> K^{\frac{2}{3}} H^{\frac{5}{3}} S^{\frac{2}{3}} A^{\frac{1}{3}}\right)$. We have updated this in our updated manuscript.
>
> # 2. Confusing step in Lemma 4
> We thank the reviewer for pointing out a confusing step in the proof, and we have corrected this in our updated manuscript. We add the discussion of the constraint of  $\tilde{\eta}$.  We remark that Lemma 4 essentially allows us to transform the constrained optimization problem $\sigma_P(V)$ to Eq.3, which is unconstrained due to several steps of optimizing the dual variables out. This is done through a change of variable from $\eta$ to $\tilde{\eta}$. Despite the constraint $\tilde{\eta} - \min_s \hat{V}(s) \leq \lambda$, one can show that $\tilde{\eta}$ still takes range in $\mathbb{R}$. This thus allows us to derive the optimal $\lambda$ in pg.16. The resultant optimization problem can then be computed rather efficiently compared to the original problem $\sigma_P(V)$.
>
> # 3. Result of Badrinath \& Kalathil (2021)
> The results in Badrinath \& Kalathil (2021) are indeed asymptotic, we have corrected this in our updated manuscript.
>
> # 4. Experiments with $s$-rectangular set
> We have updated the $s$-rectangular uncertainty sets results in the manuscript (Experiment section and Appendix F). The testing environment is created by adding a random perturbation in the direction against the optimal direction (which is towards the right-down goal state). We will make the experiments (code, environment, and logs) openly available upon acceptance of this work.
>
>  ## Reference
>
> [1] Wiesemann, Wolfram, Daniel Kuhn, and Berç Rustem. "Robust Markov decision processes." Mathematics of Operations Research 38.1 (2013): 153-183.
>
> [2] Dann, Christoph, Tor Lattimore, and Emma Brunskill. "Unifying PAC and regret: Uniform PAC bounds for episodic reinforcement learning." Advances in Neural Information Processing Systems 30 (2017).

---

> ### Author Response · Authors · 2022-11-08
> **Response to Reviewer yGXw (part 2)**
>
> # 5. Response to other comments
> * The policy optimization step in Algorithm 1 should be without the $-$, we have corrected this in the updated manuscript. This is a result of taking $\arg\max$ in the policy improvement step.
> * We thank the reviewer for pointing out a typo of  "By Lemma 2 and Lemma 3" in our proof and we have corrected this in our updated manuscript.
> * The overall simplex set for the uncertainty set under $(s,a)$-rectangular assumption is $\Delta(SAH)$. However, we defined the uncertainty set individually for each $s,a,h$ in Definition 3.1. The individual simplex set is thus $\Delta(S)$.
> * The sentence "a fixed step of time-dependent uncertainty kernels" is a typo, we were referring to ``a fixed time-dependent uncertainty kernel''.
> * We thank the reviewer for pointing out that the definition of $\Delta(S)$ in section 3 is not provided. and we have added this to our updated manuscript.
> * We say that the $s$-rectangular set (Definition 3.2) is more general as it includes the case of $(s,a)$-rectangular set. Compared to the $(s,a)$-rectangular set, where the adversarial perturbation on the transition has to be independent for each $(s,a)$ pair, the $s$-rectangular set only requires the perturbation to be independent for each state. As a result, the solution under $s$-rectangular uncertainty set may still be a solution to $(s,a)$-rectangular set, but not vice versa. We refer to [1] for a more detailed discussion of the difference between the two uncertainty sets.
> * The robust MDP is only known to be NP-hard if the agent interacts with an arbitrarily chosen transition or if the uncertainty set is arbitrary. Under Definitions 3.1 and 3.2, the problem is known to be solvable in polynomial time. We refer to Table 1 of [1] for a detailed discussion of the complexity with respect to different uncertainty sets.
> We thank the reviewer for pointing out the typo in the estimator of the transition and we have corrected this in our updated manuscript.
>
> * We thank the reviewer for pointing out that the definition of $\hat{V}$ in section 4 is not defined and we have added this in our updated manuscript.
>
> * We thank the reviewer for pointing out that the value function used in Eq.4 should be the estimator of value function $\hat{V}$ and we have corrected this in our updated manuscript.
>
> * Statement above ``Policy Improvement Step'': We have corrected the typo, we were referring to the fact that $\rho$ and $\hat{P}_h^o$ are now in different terms as a result of eq.3 and eq.4. This allows us to decouple the uncertainty in estimation error in robustness.
>
> * We thank the reviewer for pointing out the typo of 'argmax' and we have corrected this in our updated manuscript.
>
> * Gradient of $\hat{V}^{\pi_k}$: Since the value function $V^\pi$ is defined as $V^\pi(s) = \mathbb{E}_{a \sim\pi}\left[Q^\pi(s,a)\right] = \langle Q^\pi(s,\cdot),\pi(\cdot \mid s) \rangle$, the gradient of $\hat{V}^{\pi_k}$ is the Q-value function.
>
> ## Reference
> [1] Wiesemann, Wolfram, Daniel Kuhn, and Berç Rustem. "Robust Markov decision processes." Mathematics of Operations Research 38.1 (2013): 153-183.
>
>
> [2] Dann, Christoph, Tor Lattimore, and Emma Brunskill. "Unifying PAC and regret: Uniform PAC bounds for episodic reinforcement learning." Advances in Neural Information Processing Systems 30 (2017).

---

### Decision · Program_Chairs · 2023-01-20

**Decision:**

Reject

**Justification For Why Not Higher Score:**

The paper does not yet meet the bar for publication. In particular, the paper must be revised to clarify the concerns about the theory and relation to existing work.

**Justification For Why Not Lower Score:**

n/a

**Metareview: Summary, Strengths And Weaknesses:**

The submitted paper considers the problem of learning a robust policy for MDPs with uncertainty about the decision dynamics. More specifically, they consider an online setting and are interested in deriving regret bounds for this setting. They derive theoretical results in that regard and relate it to results from the literature. Furthermore, they provide proof-of-concept type experiments illustrating that their proposed algorithm can indeed effectively learn a robust policy in 5x5 grid world environment. The paper clearly addresses an important problem, derives theoretical results that could be promising, and illustrate that their algorithm can perform beneficial to a sensible baseline. Unfortunately, there are several concerns about basic definitions (e.g., uncertainty sets and the corresponding Bellman equations) and the relation into to existing results (whether their results are correctly represented as there were several inconsistencies in the discussion between reviewers and authors).
Overall, the idea of the paper is good, the taken approach probably as well, but the paper needs to be carefully revised to clarify/correct the concerns raised in the reviews. Therefore, at this stage, the paper is not ready for publication but the authors are encouraged to improve their paper based on the reviews and submit a revised and improved version to a future venue.

**Summary Of Ac-Reviewer Meeting:**

Reviewers QpYY and JsyR and the AC had a virtual discussion. Reviewers yGXw and ZEUJ could not participate in this meeting.

The concerns of the reviewers are precisely as stated in the meta review. The authors failed to clarify the concerns despite a quite active discussion with the reviewers. Hence, the paper should not be accepted as is.